# Effect of Sex on the Association Between Nonmedical Use of Opioids and Sleep Disturbance Among Chinese Adolescents: A Cross-Sectional Study

**DOI:** 10.3390/ijerph16224339

**Published:** 2019-11-07

**Authors:** Di Xiao, Lan Guo, Meijun Zhao, Sheng Zhang, Wenyan Li, Wei-Hong Zhang, Ciyong Lu

**Affiliations:** 1Department of Medical Statistics and Epidemiology, School of Public Health, Sun Yat-sen University, Guangzhou 510080, China; Di.Xiao@UGent.be (D.X.); guolan3@mail.sysu.edu.cn (L.G.); zhaomj5@mail2.sysu.edu.cn (M.Z.); zhangsh46@mail2.sysu.edu.cn (S.Z.); liwy23@mail2.sysu.edu.cn (W.L.); 2International Centre for Reproductive Health (ICRH), Department of Public Health and Primary Care, Ghent University, 9000 Gent, Belgium; 3Guangdong Provincial Key Laboratory of Food, Nutrition and Health, Sun Yat-sen University, Guangzhou 510080, China; 4Research Laboratory for Human Reproduction, Faculty of Medicine, Université Libre de Bruxelles (ULB), 1050 Bruxelles, Belgium

**Keywords:** non-medical prescription opioid use, sleep disturbance, sex differences, adolescents

## Abstract

Sleep disturbance and non-medical prescription opioid use (NMPOU) are currently growing public health concerns, and sex differences may result in differential exposure to frequency of NMPOU or sleep disturbance. This study aimed to explore the association between the frequency of lifetime or past-year NMPOU and sleep disturbance and to evaluate whether there was any sex difference in this association among Chinese adolescents. A cross-sectional study was performed in seven randomly selected Chinese provinces through the 2015 School-Based Chinese Adolescents Health Survey. A total of 159,640 adolescents were invited to participate and among them, 148,687 adolescents’ questionnaires were completed and qualified for this study (response rate: 93.14%). All analyses were performed for boys and girls separately. There were significant sex differences in the prevalence of lifetime or past-year opioid misuse and sleep disturbance (*p* < 0.05). Among girls, frequent lifetime NMPOU (adjusted odds ratio [aOR] = 2.09, 95% CI = 1.80–2.44) and past-year NMPOU (aOR = 2.16, 95% CI = 1.68–2.77) were positively associated with sleep disturbance. Among boys, these associations were also statistically significant, while the magnitudes of associations between frequent lifetime NMPOU or past-year NMPOU and sleep disturbance were greater in girls than those in boys. There is a significant sex difference in the prevalence of lifetime or past-year NMPOU and sleep disturbance. Furthermore, exposure to more frequent lifetime or past-year NMPOU is associated with a greater risk of sleep disturbance, especially among girls. Taking into account the sex difference for lifetime or past-year NMPOU may help to decrease the risk of sleep disturbance.

## 1. Introduction

Sleep plays an important role in the development and maintenance of physical and mental health, especially in adolescents [1,2]. Evidence shows that sleep disruption is highly prevalent among adolescents [3]. Research in western countries has shown that approximately 25%–40% of adolescents have sleep disorders [4,5]. In China, it was estimated that the prevalence rate of sleep disturbance among adolescents ranged from 18.0% to 39.6% [6,7,8]. Recent studies have shown that sleep problems were associated with serious behavioral problems [9], depression [10], Attention-Deficit/Hyperactivity Disorder (ADHD) [11], poor school performance [12], and suicide [13] among children and adolescents. Sleep disturbance among adolescents has become a major international health concern.

The rise in non-medical prescription opioid use (NMPOU) is an emerging public health problem in adolescents [14]. Over the past two decades, the prevalence of prescription opioid misuse has increased more than threefold in the United States [15]. Most recent evidence from the National Survey on Drug Use and Health (NSDUH) showed that 3.5% of adolescents aged 12 to 17 engaged in NMPOU in the past year [16]. Moreover, based on the data from the 2014 NSDUH, 7.3% of responding adolescents aged 12–17 reported NMPOU during their lifetime [17]. A survey in China also found that NMPOU was prevalent among Chinese adolescents, and the prevalence of the lifetime NMPOU use was 7.7% [18]. In China, the most commonly used medicine was licorice tablets with morphine, cough syrup with codeine, diphenoxylate, and tramadol [6]. Further, Guo et al. [19] reported that “to relax or relieve tension” was the most prevalent reason for nonmedical use of opioids among adolescents. The misuse of prescription opioids can lead to numerous adverse consequences, such as sexual behavior [20], heroin use [21], drug injection [21], depression [22], and suicidal behavior [23]. Evidence has shown that opioids can increase wakefulness and decrease total sleep time [24], delta sleep [25], sleep efficiency [25], and rapid eye movement (REM) sleep [26]. Walker et al. [27] reported a dose-dependent relationship between chronic opioid use and sleep disorders. Although there have been indications that past-month NMPOU could predict poor sleep [6], there is a dearth of studies addressing the potential association between the frequency of lifetime or past-year NMPOU and sleep disturbance in adolescents.

Sex differences may lead to differential exposure to prescription drug misuse or sleep disturbance. The prevalence of NMPOU in boys is reportedly greater than in girls [28,29], and girls complain more about their sleep than boys [30]. Research found that boys may have a stronger need for sensation-seeking than girls [31], and seeking pleasure and release of tension were the most common reasons for substance use [32,33]. Moreover, differences in the process of physiological changes related to puberty may lead to the sex differences in sleep disorders [34]. Whether the effects of lifetime or past-year NMPOU on sleep disturbance are similar or different for boys and girls is not clear and little attention has been paid to sex differences across these associations. Understanding the relationship between NMPOU and sleep disturbance among adolescents, as well as identifying the mechanisms that underlie potential sex differences may provide valuable insights and facilitate the design of sex-sensitive sleep disturbance preventive programs.

Therefore, to address the above questions, a large national study in China was conducted to assess the prevalence of lifetime NMPOU, past-year NMPOU, and sleep disturbance among Chinese adolescents; to explore the independent associations between the frequency of lifetime and past-year NMPOU with sleep disturbance; and to investigate whether there are sex differences within the associations.

## 2. Materials and Methods

### 2.1. Study Design

The cross-sectional data of the current study were collected from the 2015 School-based Chinese Adolescents Health Survey (SCAHS) [18,35]. SCAHS is an ongoing survey of health-related behaviors in 7th–12th grades students in China, which has been conducted every two years since 2007. The 2015 SCAHS is the most recent version [18].

### 2.2. Data Collection and Sample

Data collection procedures have been described in detail elsewhere [36]. Briefly, adolescents were selected using a multistage, stratified cluster sampling method. In stage 1, based on geographic location, a total of seven large provinces in China were selected. Each province was divided into three stratifications according to the Gross Domestic Product (GDP: high, medium, and low), then two cities were selected randomly from each stratus. In stage 2, from each representative city, the schools were classified according to three categories, including junior high school (grades 7–9), senior high school (grades 10–12), and vocational school (grades 10–12). We randomly selected 6 to 7 junior high schools, 4 to 5 senior high schools, and 2 to 3 vocational schools from each city. In stage 3, we randomly selected two classes from each grade and we investigated all the available students in these classes. Finally, a total of 159,640 adolescents were invited to participate in our survey, and 148,687 Chinese students completed the questionnaires (response rate: 93.14%). To avoid any potential information bias, a Chinese-language self-administered questionnaire was completed by each student within one class period with the supervision of research assistants. To protect the privacy of the students, the questionnaire was completed anonymously and without a teacher present.

### 2.3. Measures

#### 2.3.1. Dependent Variable

Sleep quality and disturbances over the past month were assessed by the Chinese version of the Pittsburgh Sleep Quality Index (PSQI). The Chinese version of the PSQI has been validated [37] and has been extensively used [8,38] with Chinese adolescents. The survey assessed the 19-item PSQI, which consists of seven components containing subjective sleep quality, sleep latency, sleep duration, habitual sleep efficiency, sleep disturbance, use of sleep medications, and daytime dysfunction. The sum of the scores for these seven components yields one global score with a range from 0 to 21, with higher scores indicating a higher level of sleep disturbance [37]. In China, a PSQI global score that is greater than 7 points indicates poor sleep quality, which is collectively known as sleep disturbance [7].

#### 2.3.2. Independent Variable

In the present study, four opioid drugs were investigated: cough syrup compounded with codeine, compounded licorice tablets (opium), tramadol hydrochloride, and diphenoxylate. Lifetime NMPOU was measured by the following question: “Have you ever used the following list of prescription opioid drugs even once, when you were not sick or just for the intended purpose to experiment or to get high without a doctor’s prescription?” The question was followed by a list of the above prescription opioid drugs, with responses coded as “never = 0”, “once or twice” = 1, and “at least three times” = 2. If the answer was “once or twice” or “at least three times”, we then asked about the student’s past-year NMPOU. Students who reported “never” were classified as abstainers, those who admitted once or twice were classified as experimenters, and those students who answered at least three times were classified as frequent users [19].

#### 2.3.3. Other Variables

The demographic variables included age, sex (boy = 1 and girl = 2), grade, academic pressure, academic achievement, classmate relationships, relationships with teachers, bullying experience, current smoking, and current drinking. Academic pressure and academic achievement were assessed by asking about the student’s self-rating of his or her academic pressure or achievement relative to that of his or her classmates (responses were coded as “below average”, “average”, or “above average”). Relationships with teachers and classmate relationships were measured according to the students’ self-rating of their relationships with their teachers and classmates (categorized into “good = 1”, “average = 2”, and “poor = 3”).

Bullying experience was measured with the Olweus Bully/Victim Questionnaire. The respondents were asked about the following question: “How often have you been bullied (kicked, intentionally excluded from participating, made fun of with sexual jokes, etc.) at school in the past 30 days?” [39]. The response options were (1) “never”, (2) “sometimes or rarely (one or two times)”, or (3) “often (more than three times)”. Students who selected a frequency of “often” in the past 30 days were defined as being bullied [40].

Current smoking was investigated by asking students the following question: “How many days did you smoke cigarettes during the past 30 days?” Responses were defined as current smokers when the selected answers indicated 1 or more days [41]. Current drinking was assessed with the following question: “During the past 30 days, on how many days did you drink alcohol?” Responses were defined as current drinkers when the selected answers indicated 1 or more days [42].

### 2.4. Ethical Considerations

The Sun Yat-Sen University School of Public Health Institutional Review Board approved this study (L2014076). Then, a written informed consent form was obtained from each school and one of the parents (or legal guardians) of each participating adolescent after the study procedures had been fully described.

### 2.5. Statistical Analysis

SAS 9.4 (SAS Institute, Inc., Cary, NC, USA) was used to perform all statistical analyses. Firstly, to describe the sample characteristics, the prevalence of lifetime or past-year NMPOU, and sleep disturbance, descriptive analyses that were stratified by sex were conducted. Secondly, to investigate whether there were any statistically significant differences between female and male students, a t-test for continuous variables and chi-square test for categorical variables were performed. Thirdly, univariable logistic regression models were conducted to test the potential factors that are associated with sleep disturbance. Finally, multivariable logistic regression models were used to explore the independent associations of lifetime and past-year NMPOU with sleep disturbance, and the covariates that were associated (*p* value < 0.05) with sleep disturbance in univariable analyses were entered simultaneously as control variables.

Statistical significance was set at the *p* < 0.05 using two-sided tests.

## 3. Results

### 3.1. Population Characteristics by Sex

Table 1 presents the basic demographic information of this study. Of the 148,687 students, 48.0% (71,442) were boys and 52.0% (77,245) were girls. The age range was 12–18 years and the mean age of the students was 15.0 (SD: ±1.8). Overall, 1.8% and 0.7% of the students admitted that they were a “frequent user” of lifetime or past-year NMPOU, respectively. Among the total sample, 35.7% of the students rated their academic pressures as above average and 8.2% of the students had bullying experience. Approximately 5.3% of adolescents reported current smoking and 15.9% of the students currently drank alcohol. A total of 21.6% students reported having sleep disturbance. Statistically significant differences were observed in the sex distribution of age, grade, academic achievement, teacher–classmate and classmate relations, academic pressure, bullying experience, current smoking, current drinking, lifetime, and past-year NMPOU (*p* < 0.05).

### 3.2. Prevalence of Sleep Disturbance by Sex

Among boys and girls, the prevalence of sleep disturbance was 20.6% and 22.5%, respectively (Table 1). Without adjusting for other variables, sleep disturbance was more prevalent among adolescents who were frequent lifetime NMPOU (36.2% among boys versus 41.8% among girls) and those with frequent past-year NMPOU (40.9% among boys versus 47.0% among girls). Among both boys and girls, lifetime and past-year NMPOU, age, grade, academic achievement, teacher–classmate, classmate relations, academic pressure, bullying experience, current smoking, and current drinking were associated with sleep disturbance (Table 2).

### 3.3. Association Between Lifetime or Past-Year NMPOU and Sleep Disturbance

Table 3 and Figure 1 show the results of the final multivariable logistic regression models for sleep disturbance. After adjusting for age, grade, academic achievement, teacher–classmate and classmate relations, academic pressure, bullying experience, current smoking, and current drinking, lifetime and past-year NMPOU were significantly associated with sleep disturbance among both boys and girls. Among boys, frequent lifetime and past-year NMPOU was significantly associated with sleep disturbance, with aORs of 1.80 (95% CI, 1.57–2.05) and 2.07 (95%CI, 1.66–2.58), respectively. Among girls, frequent lifetime and past-year NMPOU were also positively associated with sleep disturbance, with aORs of 2.09 (95% CI, 1.80–2.44) and 2.16 (95%CI, 1.68–2.77), respectively. These results indicated that the associations between frequent lifetime and past-year NMPOU and sleep disturbance were higher among girls than boys. Moreover, the magnitudes of aORs for the significant associations between frequent lifetime or past-year NMPOU and sleep disturbance were greater than those between experimental use and sleep disturbance among both boys and girls.

## 4. Discussion

### 4.1. Main Findings

This is the large-scale study aimed to investigate the influence of lifetime or past year NMPOU on adolescents’ sleep disturbance. The present results demonstrate that 21.6% of the sampled students reported having sleep disturbance. This prevalence is higher than that measured in a study conducted in Hefei of China, which indicated that 18.6% adolescents admitted having sleep disturbance [43]. To date, the role of sex for sleep disturbance among adolescents is not well characterized. Moreover, data from individual studies on the sex differences in sleep disturbance are not entirely consistent [44,45,46]. For example, Hartley et al. [44] found that girls reported more sleeping problems than did boys, while Russo et al. [45] reported the opposite results. Moreover, Morrison et al. [46] found that there were no significant differences between boys and girls regarding their number of sleep problems. With the large sample, our results demonstrate a higher prevalence of sleep disturbance among girls compared to boys.

Our results also illustrated that lifetime NMPOU was prevalent among Chinese adolescents, and the prevalence was higher for boys than girls. Furthermore, this study indicated that boys were found to have higher prevalence rates of experimental and frequent use of opioids in their lifetime or the past year. This finding is consistent with a previous study, which found that nonmedical use of opioids was more prevalent in boys than girls [47]. Some evidence also supported that males generally reported more substance use than females [48]. The observed sex differences in susceptibility to nonmedical opioids use may be useful to enhance the design of sex-sensitive surveillance, identification, prevention, and treatment decisions toward NMPOU [49].

The univariate analyses found that younger girls were more likely to have sleep disturbances. Previous studies also suggested that girls’ sleep problems are usually manifested at an earlier age than boys [30,50]. Different social requirements for boys and girls and differences in the process of physiological changes linked to puberty might account for the difference [30,51]. In addition, we observed that adolescents who reported having bullying experience were at a higher risk of sleep disturbance. Similarly, Zhou et al. [8] found that adolescents who were victims of bullying had a twofold increased risk of having sleep disturbances. Unsurprisingly, compared to their corresponding groups, students who reported current smoking or drinking had a higher risk of having sleep disturbances. Higher grades (10th to 12th), poor relationships with teachers or classmates, and higher academic pressure were positively associated with higher risk for sleep disturbance. In light of the growth of sleep disturbance and the host of adverse consequences that are linked with it, it is critical to recognize the high-risk students who will be more prone to sleep disturbance. Specifically, sex, age, grades, bullying experience, current smoking or drinking, relationships with teachers or classmates, academic pressure and achievement were showed to be differentially associated with sleep disturbance among adolescents. Gaining a better understanding of the nuances of sleep disturbance trends will support more refined health promotion efforts. Therefore, we suggested that clinicians, schools, and families should pay more attention to these high-risk students mentioned above with the adverse characteristics to reduce the potential risk of sleep disturbance.

Our findings provide some evidence that compared with experimental opioid misuse, more frequent prescription opioid misuse was associated with a greater likelihood of reporting sleep disturbance after adjusting for covariates. We speculated that compared to experimental users of opioids, individuals who reported more frequent opioid misuse are more likely to progress to dependent users who experience more interpersonal conflict, which then can contribute to triggering negative effects on physical health, ultimately leading to sleep disturbance [52].

Notably, another novel discovery of the study was the effects of the separate analyses by sex, which suggested that the adjusted associations between lifetime or past-year NMPOU and sleep disturbance appeared slightly higher in girls than boys. A possible explanation is that females who reported opioids misuse were found to have greater levels of global psychiatric and emotional distress [53,54], and emotional trouble can lead to difficulty falling asleep in addition to further sleep disturbance [55].

### 4.2. Limitations

The limitations of the present study should be acknowledged when interpreting these findings. Firstly, it should be noted that a previous study found a bivariate relationship between the nonmedical use of prescription drugs and sleep among adolescents [56]. Furthermore, given the cross-sectional nature of the present study, our data cannot determine the exact timeline between lifetime or past-year NMPOU and sleep disturbance. Therefore, these issues should be further explored in a longitudinal study. Secondly, the data only included adolescents who were attending school, and on the day the survey was administered, adolescents who were not present at school or had dropped out of school were excluded. However, the anonymity of the questionnaires is assured and this method may have helped to collect accurate information from adolescents. Moreover, there could also be other potential confounders that were not included, such as mood disorders. Future surveys could collect the related information. Finally, our data only investigated the most widely used opioid drugs of Chinese adolescents. Hence, more prescription drugs need to be explored in future research.

Despite these limitations, the strength of the findings is that a large-scale sample of Chinese students was investigated, which ensures sufficient statistical power to measure the possible associations between lifetime or past-year NMPOU and sleep disturbance after adjusting for control variables. Besides, to our knowledge, the effect of sex on the association of lifetime or past-year NMPOU with sleep disturbance has not specifically been reported before. This is the first study that aimed to determine a sex-differential association among Chinese adolescents.

### 4.3. Implications

In China, NMPOU and sleep disturbance are both ongoing and rapidly evolving public health concerns. The findings clearly suggest that girls have a greater prevalence of sleep disturbance compared to boys and boys have a higher prevalence of experimental and frequent use of opioids in their lifetime and the past-year than girls. Furthermore, the study finds that although both boys and girls who reported lifetime or past-year NMPOU are at an increased risk of sleep disturbance, the association between frequency of opioid misuse and sleep disturbance is slightly stronger in girls. These results expand the areas of the literature, providing more specific information regarding the association between NMPOU and sleep disturbance. Also, the findings of this study highlight clinically relevant sex differences and may be helpful in guiding the design of sex-sensitive screening and treatment interventions for sleep disturbance [54]. Specific prevention and intervention programs are suggested: (1) to educate adolescents about avoiding the use of opioids as a way to deal with stress [57]; (2) to improve the awareness of the negative outcomes of NMPOU and sleep disturbance by educational campaigns; (3) to build a nation-wide active monitoring system to supervise NMPOU, sleep disturbance, and other health-risk behavior (e.g., smoking, drinking alcohol, and bulling victimization, etc.) among Chinese adolescents.

## 5. Conclusions

To conclude, this report identified sex differences in the prevalence of lifetime or past-year NMPOU. Moreover, exposure to more frequent lifetime or past-year NMPOU is associated with a greater risk of sleep disturbance, especially among girls. Future intervention or prevention strategies should take into account the sex difference on the effects of sleep disturbance among adolescent nonmedical-opioid users in China.

## Figures and Tables

**Figure 1 ijerph-16-04339-f001:**
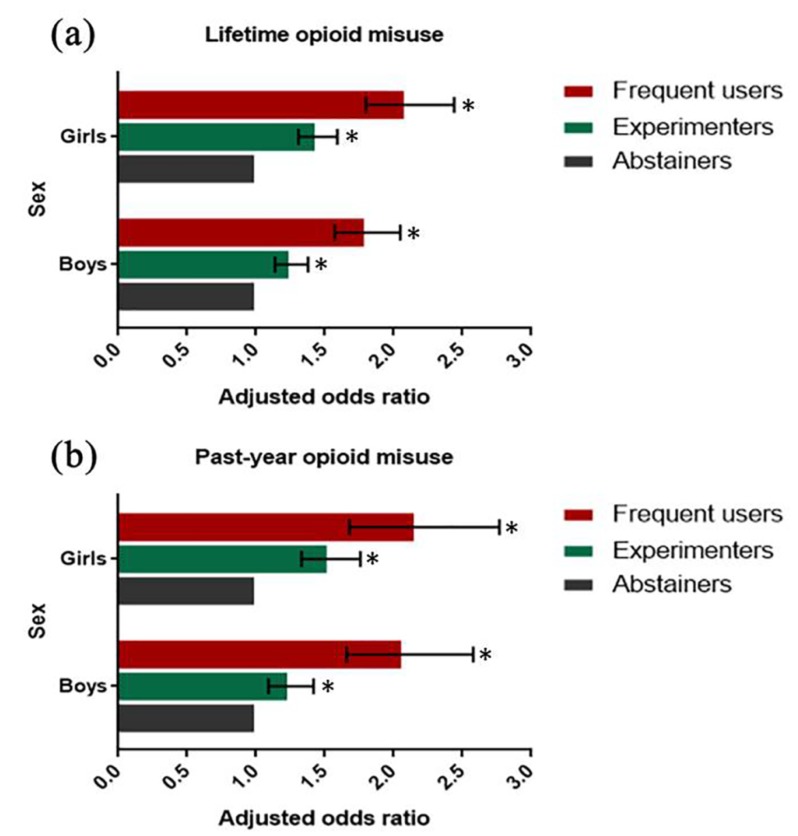
Adjusted OR and 95%CI for measuring the association of lifetime or past-year NMPOU with sleep disturbance stratified by sex.* *p* < 0.05.

**Table 1 ijerph-16-04339-t001:** Sex difference of the sample characteristics (*n* = 148,687).

Variable	Total, No. (%)	Boys	Girls	*p*-Value ^1^
**Total**	148,687	71,442 (48.0)	77,245 (52.0)	<0.001
**Age**				
12 to 13	34,890 (23.5)	16,844 (23.6)	18,046 (23.4)	<0.001
14 to 15	51,225 (34.5)	25,782 (36.1)	25,443 (32.9)	
16 to 18	62,572 (42.1)	28,816 (40.3)	33,756 (43.7)	
**Grade**				
7th to 9th	77,936 (52.4)	39,717 (55.6)	38,219 (49.5)	<0.001
10th to 12th	70,751 (47.6)	31,725 (44.4)	39,026 (50.5)	
**Academic achievement ^596^**				
Above average	52,839 (35.7)	24,364 (34.2)	28,475 (37.0)	<0.001
Average	49,782 (33.6)	21,172 (29.7)	28,610 (37.2)	
Below average	45,470 (30.7)	25,651 (36.0)	19,819 (25.8)	
**Teacher classmate relations ^515^**				
Good	81,130 (54. 8)	39,494 (55.5)	41,636 (54.1)	<0.001
Average	61,886 (41.8)	28,094 (39.5)	33,792 (43.9)	
Poor	5156 (3.5)	3564 (5.0)	1592 (2.1)	
**Classmate relations ^465^**				
Good	108,178 (73.0)	52,656 (74.0)	55,522 (72.1)	<0.001
Average	37,389 (25.2)	16,873 (23.7)	20,516 (26.6)	
Poor	2655 (1.8)	1664 (2.3)	991 (1.3)	
**Academic pressure ^182^**				
Below average	23,238 (15.6)	12,712 (17.8)	10,526 (13.6)	<0.001
Average	68,591 (49.2)	31,494 (44.1)	37,097 (48.1)	
Above average	56,676 (38.2)	27,147 (38.0)	29,529 (38.3)	
**Bullying experience**				
No	136,458 (91.8)	63,204 (88.5)	73,254 (94.8)	<0.001
Yes	12,229 (8.2)	8238 (11.5)	3991 (5.2)	
**Current smoking**				
No	140,855 (94.7)	64,708 (90.6)	76,147 (98.6)	<0.001
Yes	7832 (5.3)	6734 (9.4)	1098 (1.4)	
**Current drinking**				
No	125,120 (84.1)	56,357 (78.9)	68,763 (89.0)	<0.001
Yes	23,567 (15.9)	15,085 (21.1)	8482 (11.0)	
**Lifetime NMPOU**				<0.001
Abstainers	139,321 (93.7)	66,426 (93.0)	72,895 (94.4)	
Experimenters	6679 (4.5)	3529 (4.9)	3150 (4.1)	
Frequent users	2687 (1.8)	1487 (2.1)	1200 (1.6)	
**Past-year NMPOU**				<0.001
Abstainers	144,447 (97.1)	69,139 (96.8)	75,308 (97.5)	
Experimenters	3228 (2.2)	1738 (2.4)	1490 (1.9)	
Frequent users	1012 (0.7)	565 (0.8)	447 (0.6)	
**Sleep disturbance**				
No	116,522 (78.4)	56,694 (79.4)	59,828 (77.5)	<0.001
Yes	32,165 (21.6)	14,748 (20.6)	17,417 (22.5)	

NMPOU, non-medical prescription opioid use; Number of missing data were listed in superscript; ^1^ Chi-square tests were used to test the association between the above-mentioned categories and sex.

**Table 2 ijerph-16-04339-t002:** Lifetime prevalence, crude odds ratios, and 95% CI of sleep disturbance among adolescents: stratified by sex.

Variable	Sleep Disturbance				
	Total *n* (%)	Boys *n* (%)	cOR (95%CI)	Girls *n* (%)	cOR (95%CI)
**Lifetime NMPOU**					
Abstainers	29,153 (20.9)	13,263 (20.0)	1	15,890 (21.8)	1
Experimenters	1972 (29.5)	946 (26.8)	1.49 (1.36–1.62)	1026 (32.6)	1.71 (1.57–1.87)
Frequent users	1040 (38.7)	539 (36.2)	2.25 (2.00–2.54)	501 (41.8)	2.65 (2.31–3.03)
**Past-year NMPOU**					
Abstainers	30,720 (21.3)	14,033 (20.3)	1	16,687 (22.2)	1
Experimenters	1004 (31.1)	484 (27.8)	1.50 (1.33–1.70)	520 (34.9)	1.90 (1.67–2.16)
Frequent users	441 (43.6)	231 (40.9)	2.74 (2.26–3.33)	210 (47.0)	2.99 (2.43–3.67)
**Age**					
12 to 13	3763 (10.8)	1723 (10.2)	1	2040 (11.3)	1
14 to 15	9838 (19.2)	4480 (17.4)	1.81 (1.67–1.95)	5358 (21.1)	2.11 (1.94–2.29)
16 to 18	18,564 (29.7)	8545 (29.7)	3.64 (3.35–3.95)	10,019 (29.7)	3.33 (3.07–3.61)
**Grade**					
7th to 9th	11,097 (14.2)	5328 (13.4)	1	5769 (15.1)	1
10th to 12th	21,068 (30.0)	9420 (30.0)	2.70 (2.54–2.88)	11,648 (30.0)	2.37 (2.21–2.53)
**Academic achievement**					
Above average	9762 (18.5)	4350 (17.9)	1	5412 (19.0)	1
Average	10,616 (21.3)	4194 (19.8)	1.15 (1.08–1.22)	6422 (22.4)	1.23 (1.17–1.29)
Below average	11,704 (25.7)	6177 (24.1)	1.48 (1.40–1.56)	5527 (27.9)	1.66 (1.57–1.74)
**Teacher–classmate relations**					
Good	13,605 (16.8)	6332 (16.0)	0.58 (0.55–0.61)	7273 (17.5)	0.55 (0.53–0.58)
Average	16,327 (26.4)	6956 (24.8)	1	9371 (27.7)	1
Poor	2145 (41.6)	1411 (39.6)	1.92 (1.75–2.10)	734 (46.1)	2.21 (1.95–2.50)
**Classmate relations**					
Good	20,772 (19.2)	9668 (18.4)	0.61 (0.58–0.64)	11,104 (20.0)	0.62 (0.59–0.65)
Average	10,288 (27.5)	4413 (26.2)	1	5875 (28.6)	1
Poor	1018 (38.3)	621(37.3)	1.64 (1.45–1.86)	397 (40.1)	1.69 (1.44–1.98)
**Academic pressure**					
Below average	2918 (12.6)	1701 (13.4)	1	1217 (11.6)	1
Average	10,706 (15.6)	4569 (14.5)	0.62 (0.55–0.69)	6137 (16.5)	0.72 (0.62–0.83)
Above average	18,517 (32.7)	8472 (31.2)	1.35 (1.23–1.48)	10,045(34.0)	1.89 (1.66–2.15)
**Bullying experience**					
No	27,600 (20.2)	11,858 (18.8)	1	15,742 (21.5)	1
Yes	4565 (37.3)	2890 (35.1)	2.38 (2.24–2.54)	1675 (42.0)	2.64 (2.44–2.85)
**Current smoking**					
No	29,223 (20.7)	12,328 (19.1)	1	16,895 (22.2)	1
Yes	2942 (37.6)	2420 (35.9)	2.39 (2.23–2.57)	522 (47.5)	3.43 (2.95–4.00)
**Current drinking**					
No	24,382 (19.5)	10,107 (17.9)	1	14,275 (20.8)	1
Yes	7783 (33.0)	4641 (30.8)	2.01 (1.90–2.12)	3142 (37.0)	2.26 (2.13–2.40)

NMPOU, non-medical prescription opioid use; cOR, crude odds ratio; 95% CI, 95% confidence interval.

**Table 3 ijerph-16-04339-t003:** Adjusted odds ratios and 95%CI for measuring the association between opioids use and sleep disturbance stratified by sex.

Variable	Sleep Disturbance	
	Boys, aOR (95%CI) ^1^	Girls, aOR (95%CI) ^1^
**Lifetime NMPOU**		
Abstainers	1	1
Experimenters	1.25 (1.14–1.38)	1.44 (1.31–1.59)
Frequent users	1.80 (1.57–2.05)	2.09 (1.80–2.44)
**Past-year NMPOU**		
Abstainers	1	1
Experimenters	1.24 (1.09–1.42)	1.53 (1.33–1.76)
Frequent users	2.07 (1.66–2.58)	2.16 (1.68–2.77)

NMPOU, non-medical prescription opioid use; aOR, adjusted odds ratio; 95% CI, 95% confidence interval. ^1^ Adjusted for age (years), grade, academic achievement, teacher–classmate, classmate relations, academic pressure, bullying experience, cigarette smoking, current drinking.

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
