# Peer review of "Effect of Sex on the Association Between Nonmedical Use of Opioids and Sleep Disturbance Among Chinese Adolescents: A Cross-Sectional Study"

_ijerph, 2019, doi:10.3390/ijerph16224339_

Round 1

Reviewer 1 Report

The manuscript “Effect of the Sex on the Association between Nonmedical Use of Opioids and Sleep Disturbance among Chinese Adolescents: A Cross-Sectional Study” addresses an important and timely question, is carefully analyzed, and cautiously interpreted. Some omissions and minor language issues are noted below.

General Concerns

If it was consistent with general policies and the survey didn’t contain any copyrighted items, consider including the actual survey as a supplemental material. As an international audience may not be familiar with China, please list the language the survey was in and a bit of bit picture context about language in China (e.g. how many spoken languages there are, how many students learn Mandarin). At least for American teenagers, some will not take a survey seriously (will enter the same response repeatedly). The US Monitoring the Future will have some internal quality checks like excluding responses where they used a substance in the past month but not lifetime. If these types of quality control strategies were employed, these should be briefly described in the methods.

Minor Points

Line 19: are currently growing public health concerns

L 20: between the frequency of

L 23: replace 7 with seven

L 24: A total of

L 34: The sentence “Take into  …” is awkward and could be revised (e.g. Taking into account … of sleep disturbances.”

L 41: Research (singular)

L 46: has become

L 49: “upsurged” is an odd phrasing, consider “increased”

L 51: from the 2014

L 52: A survey in China

L 53: was 7.7%.

L 83: , a total of seven

L 94: and without a teacher

L 97: by the Chinese

L 107: delete “(codeine)”,

L 112: The translation of certain items is clunky. “drugs past year”

L 129: Students who (selected)

L142: Consider the potential of including the data file in a common format (xls) as a supplemental material. This would improve transparency but is only a suggestion.

L 157: avoid starting a sentence with a # (e.g. 1.8%)

L 165: Capitalize Girls. Stylistically, consider use of commas for “148,687”, “34,890”, etc. It’s a lot of space that gets filled to have a row for “Missing data” that gets repeated. Consider instead listing that N in superscript after each variable (e.g. Academic achievement596 ).

L 188: Consider an * over the error bars that are statistically significant. The key should be in the same sequence as the bars (Frequent, Exp, then Abstainers).

L 236: Delete “Overview,”

L 239: Replace “And” with “A”

L 241: This evidence

L 242: our finding provides

L 243: more frequent prescription opioid

L 252: that females

L 254: The “On the other hand …” sentence is awkward. Impel?

L 263: Is it possible that students with a sleep disturbance are attempting to self-medicate with prescription opioids? If so, this should be mentioned.

L 271: which ensures

L 279: compared

L 286: Delete “And” and “as followed”

L 293: To conclude, this report identified sex differences in …

References: Please check the formatting of article titles is consistent (e.g. non-proper nouns in lower-case). Journal names should be consistently capitalized too.

Author Response

Response to Reviewer 1 Comments

Point 1: The manuscript “Effect of the Sex on the Association between Nonmedical Use of Opioids and Sleep Disturbance among Chinese Adolescents: A Cross-Sectional Study” addresses an important and timely question, is carefully analyzed, and cautiously interpreted. Some omissions and minor language issues are noted below.

Response 1: Thank you for your generous work and positive suggestions, which have definitely contributed to improving our manuscript. We have studied your comments carefully and have revised the manuscript accordingly. Our point-by-point responses to the comments are listed below.

Point 2:

Point 2.1: If it was consistent with general policies and the survey didn’t contain any copyrighted items, consider including the actual survey as a supplemental material.

Response 1: Thank you for your suggestions. Our study used the cross-sectional data collected in the 2015 School-based Chinese Adolescents Health Survey (SCAHS) [1, 2], which is an ongoing study of the health-related behaviours among Chinese adolescents. And SCAHS collects large-scale cross-sectional data (conducted every 2 years since 2007) [3, 4] via questionnaires administered in classrooms. As the SCAHS is ongoing, a copy of the questionnaire is available from the corresponding author upon request.

Point 2.2: As an international audience may not be familiar with China, please list the language the survey was in and a bit of bit picture context about language in China (e.g. how many spoken languages there are, how many students learn Mandarin).

Response 2.2: Thank you for your suggestions. In our survey, this large-scale cross-sectional data was collected through the Chinese-language self-administered questionnaire. Mandarin is national language and common teaching language in China. According to your suggestion, we have added the language of the survey (please see lines 99). Furthermore, all students participated in our survey learn our native language (Mandarin) in school. The questionnaires were administered by research assistants in the classrooms without the presence of the teachers. It means that if the participant has any question about the questionnaire, they can ask the research assistants anytime. And research assistants will explain the items in understandable terms in Mandarin till the students understand.

Point 2.3: At least for American teenagers, some will not take a survey seriously (will enter the same response repeatedly). The US Monitoring the Future will have some internal quality checks like excluding responses where they used a substance in the past month but not lifetime. If these types of quality control strategies were employed, these should be briefly described in the methods.

Response 2.3: Thank you for your suggestions. In our survey, lifetime NMPOU was assessed by the following question: “Have you ever used the following list of prescription opioid drugs even once, when you were not sick or just for the intended purpose to experiment or to get high without a doctor’s prescription?” And the question was followed by a list of above prescription opioid drugs, with responses coded as never=0, once or twice=1 and at least three times=2. If the answer was “once or twice” or “at least three times”, we then asked about the student’s past-year NMPOU (please see lines118). To avoid any potential information bias, a Chinese-language self-administered questionnaire was completed by each student within one class period with the supervision of research assistant (please see lines 100). In addition, these well trained research assistant will check every questionnaire to ensure the quality of the questionnaire in the survey. If someone completed the questionnaire obviously not serious, for example, responses which they used a substance in the past month but not lifetime. At that time, the research assistant will remind them to review the questions and answer them carefully. And the worse quality of questionnaire would not be included in the study.

Point 3: Line 19: are currently growing public health concerns

Response 3: Thank you for your correction. We have replaced “concern” with “concerns” (please see lines 19).

Point 4: L 20: between the frequency of

Response 4: Thank you. We have replaced “between frequency of” with “between the frequency of”(please see lines 21).

Point 5: L 23: replace 7 with seven

Response 5 Thank you. We have replaced “7” with “seven” (please see lines 23).

Point 6: L 24: A total of

Response 6: Thank you for your correction. We have replaced “Total of” with “A total of” (please see lines 24).

Point 7: L34 The sentence “Take into …” is awkward and could be revised (e.g. Taking into account … of sleep disturbances.”

Response 7: Thank you for your kind suggestion. We have replaced “Take into” with “Taking into account” (please see lines 35).

Point 8: L 41: Research (singular)

Response 8: Thank you. We have replaced “Researches” with “Research” (please see lines 42).

Point 9: L 46: has become

Response 9: Thank you. We have replaced “have become” with “has become” (please see lines 47).

Point 10: L 49: “upsurged” is an odd phrasing, consider “increased”

Response 10: Thank you for your kind suggestion. We have replaced “upsurged” with “has increased” (please see lines 50).

Point 11: L 51: from the 2014

Response 11: Thank you. We have replaced “from 2014” with “from the 2014” (please see lines 52).

Point 12: L 52: A survey in China

Response 12: Thank you. We have replaced “A survey of China” with “A survey in China” (please see lines 53).

Point 13: L 53: was 7.7%.

Response 13: Thank you for your correction. We have replaced “were 7.7%” with “was 7.7%” (please see lines 55).

Point 14: L 83: , a total of seven

Response 14: Thank you for your kind suggestion. We have replaced “total of 7” with “a total of seven” (please see lines 90).

Point 15: L 94: and without a teacher

Response 15: Thank you. We have replaced “and without teacher” with “and without a teacher” (please see lines 101).

Point 16: L 97: by the Chinese

Response 16: Thank you. We have replaced “by The Chinese” with “by the Chinese” (please see lines 105).

Point 17: L 107: delete “(codeine)”,

Response 17: Thank you. We have deleted “(codeine)” (please see lines 115).

Point 18: L 112: The translation of certain items is clunky. “drugs past year”

Response 18: Thank you for your kind suggestion. We have revised this sentence (please see lines 118).

Point 19: L 129: Students who (selected)

Response 19: Thank you for your kind suggestion. We have replaced “Students who” with ‘Students who selected’ (please see lines 137).

Point 20: L142: Consider the potential of including the data file in a common format (xls) as a supplemental material. This would improve transparency but is only a suggestion.

Response 20: Thank you for your kind suggestion. Data is available from the corresponding author upon request.

Point 21: L 157: avoid starting a sentence with a # (e.g. 1.8%)

Response 21: Thank you for your kind suggestion. We have replaced “Overall” with “1.8%” in starting a sentence (please see lines 165).

Point 22: L 165: Capitalize Girls. Stylistically, consider use of commas for “148,687”, “34,890”, etc. It’s a lot of space that gets filled to have a row for “Missing data” that gets repeated. Consider instead listing that N in superscript after each variable (e.g. Academic achievement596 ).

Response 22: Thank you for your kind suggestion. According to your suggestion, these numbers of missing data was listed in superscript (please see lines 173).

Point 23: L 188: Consider an *over the error bars that are statistically significant. The key should be in the same sequence as the bars (Frequent, Exp, then Abstainers).

Response 23: Thank you for your kind suggestion. We have added ‘*’ and adjusted the same sequence as the bars (Frequent, Exp, then Abstainers) in the revised Figure1 (please see lines 206).

Point 24: L 236: Delete “Overview,”

Response 24: Thank you. According to the suggestions of reviewer 3, we have deleted this sentence (please see lines 248).

Point 25: L 239: Replace “And” with “A”

Response 25: Thank you for your kind suggestion. According to the suggestions of reviewer 3, we have deleted this sentence (please see lines 248).

Point 26: L 241: This evidence

Response 26: Thank you for your kind suggestion. According to the suggestions of reviewer 3, we have deleted this sentence (please see lines 248).

Point 27: L 242: our finding provides

Response 27: Thank you for your kind suggestion. We have replaced “our finding first provides” with “our finding provides” (please see lines 248).

Point 28: L 243: more frequent prescription opioid

Response 28: Thank you for your kind suggestion. We have replaced “more frequency of prescription opioid” with “more frequent prescription opioid” (please see lines 249).

Point 29: L 252: that females

Response 29: Thank you for your kind suggestion. We have replaced “that female” with “that females” (please see lines 256).

Point 30: L 254: The “On the other hand …” sentence is awkward. Impel?

Response 30: Thank you for your kind suggestion. We have deleted this sentence “On the other hand …” (please see lines 259).

Point 31: L 263: Is it possible that students with a sleep disturbance are attempting to self-medicate with prescription opioids? If so, this should be mentioned.

Response 31: Thank you for your comments. Ayres et al. [5] reported that significant bivariate relationships were found between the nonmedical use of prescription drugs and sleep among adolescents. Therefore, longitudinal studies are needed to further explore the causal relationship between NMUPM and sleep disturbance. According to your suggestion, we have added this part in the Discussion (please see lines 262)

Point 32: L 271: which ensures

Response 32: Thank you. We have replaced “which ensure” with “which ensures” (please see lines 275).

Point 33: L 279: compared

Response 33: Thank you. We have replaced “compare” with “compared” (please see lines 283).

Point 34: L 286: Delete “And” and “as followed”

Response 34: Thank you. We have deleted “And” and “as followed” (please see lines 290).

Point 35: L 293: To conclude, this report identified sex differences in …

Response 35: Thank you for your kind suggestion. We have replaced “To conclude, this finding suggests that sex difference in” with “To conclude, this report identified sex differences in” (please see lines 297).

Point 36: References: Please check the formatting of article titles is consistent (e.g. non-proper nouns in lower-case). Journal names should be consistently capitalized too.

Response 36: Thank you for your kind suggestion. We have checked the formatting of article titles is consistent. Furthermore, journal names also be consistently capitalized in the revised reference (please see lines from 311)

Reference

Guo, L.; Xu, Y.; Deng, J.; Huang, J.; Huang, G.; Gao, X.; Wu, H.; Pan, S.; Zhang, W. H.; Lu, C., Association between nonmedical use of prescription drugs and suicidal behavior among adolescents. JAMA Pediatrics 2016, 170, 971-978. Li, P.; Huang, Y.; Guo, L.; Wang, W.; Xi, C.; Lei, Y.; Luo, M.; Pan, S.; Deng, X.; Zhang, W. H.; Lu, C., Sexual attraction and the nonmedical use of opioids and sedative drugs among Chinese adolescents. Drug and Alcohol Dependence 2018, 183, 169-175. Wang, H.; Deng, J.; Zhou, X.; Lu, C.; Huang, J.; Huang, G.; Gao, X.; He, Y., The nonmedical use of prescription medicines among high school students: a cross-sectional study in Southern China. Drug and Alcohol Dependence 2014, 141, 9-15. Guo, L.; Xu, Y.; Deng, J.; He, Y.; Gao, X.; Li, P.; Wu, H.; Zhou, J.; Lu, C., Non-medical use of prescription pain relievers among high school students in China: a multilevel analysis. Bmj Open 2015, 5, (7), e007569. Ayres, C. G.; Pontes, N. M.; Pontes, M. C. F., Understanding the nonmedical use of prescription medications in the U.S. high school adolescents. J Sch Nurs. 2017, 33, 269-276.

Reviewer 2 Report

The authors present the results of cross-sectional study to assess the prevalence of lifetime NMPOU, past-year NMPOU and sleep disturbance among Chinese adolescents, to explore the independent associations of lifetime and past-year NMPOU with sleep disturbance and to investigate whether there are sex differences in the associations.. The work is well designed, with sampling randomization.

It is not clear how did the adolecents acquired opioid drugs without a medical prescription? In pharmacies? Is it legal? The results indicatate an association between the exposure to more frequent lifetime or past-year NMPOU with a greater risk of sleep disturbance, especially among girls. However, this conclusion have to be seen carefully because some possible confounding variables were not analised, for exemple some sociodemogrpahic caracteristics already described in the literatutre as beeing associated with sleep disturbance and with opiods consumption. It will be also interesting to know ihow is the use of social drugs between the users of NMPOU.  Authors discussed some important  limitations, but it will be welcome to discuss other variables that coud influence the results.

Author Response

Response to Reviewer 2 Comments

Point 1: The authors present the results of cross-sectional study to assess the prevalence of lifetime NMPOU, past-year NMPOU and sleep disturbance among Chinese adolescents, to explore the independent associations of lifetime and past-year NMPOU with sleep disturbance and to investigate whether there are sex differences in the associations. The work is well designed, with sampling randomization.

Response 1: Thank you for your generous work and positive suggestions, which have definitely contributed to improving our manuscript. We have studied your comments carefully and have revised the manuscript accordingly. Our point-by-point responses to the comments are listed below.

Point 2: It is not clear how did the adolescents acquired opioid drugs without a medical prescription? In pharmacies? Is it legal?

Response 2: We truly thank you for your constructive comments. Our precious study showed that the nonmedically used prescription opioids or sedatives among adolescents were most commonly obtained from home (45.0%), followed by from peers (25.9%),others (21.8%) and nightclub/Pub(0.8%)[1]. And it is not legal for adolescents to acquire opioid drugs without a medical prescription.

Point 3: The results indicatate an association between the exposure to more frequent lifetime or past-year NMPOU with a greater risk of sleep disturbance, especially among girls. However, this conclusion have to be seen carefully because some possible confounding variables were not analised, for exemple some sociodemogrpahic caracteristics already described in the literatutre as beeing associated with sleep disturbance and with opiods consumption. It will be also interesting to know ihow is the use of social drugs between the users of NMPOU. Authors discussed some important  limitations, but it will be welcome to discuss other variables that coud influence the results.

Response 3: We truly thank you for your comments and constructive suggestions. We have added this limitation in the discussion (please see lines 270)

Reference

Guo, L.; Luo, M.; Wang, W.; Xiao, D.; Xi, C.; Wang, T.; Zhao, M.; Zhang, W.-H.; Lu, C., Association between nonmedical use of opioids or sedatives and suicidal behavior among Chinese adolescents: an analysis of sex differences. Australian & New Zealand Journal of Psychiatry 2019, 53, 559-569.

Reviewer 3 Report

The manuscript concerns a very significant problem in terms of mental health of adolescents. It has been properly formatted, divided into parts in accordance with the requirements of the journal. The manuscript conteins correctly selected and up-to-date literature. However, it contains a few weak points. In the introduction, the Authors should explain what opioids are abused and the reason for the increase in their use. Especially that They emphasized that adolescents use them and even abuse. Are these opioids so easily available in their country? This part of the manuscript is particularly controversial. How is it possible that even 12 year olds use them? Table 1 contains p-values but is not clearly indicated to which group the comparisions were made. This should be completed since it makes difficult to understand the results. Table 2: the description should be supplemented with the abbreviation cOR. Paragraphs 3.2 and 3.3 are rather a description of the tables than the results, this should corrected. In discussion informations in lines 236 to 241 have no direct relation to the topic. In my opinion they should be omitted. Additional manuscript weaknesses are repetitions of the same thread in discussion. The manuscript should be also supplemented with a description of the effect of opioids on sleep and sleep disorders. The authors should also indicate what is a novelty of this work, since they themselves clearly emphasize that obtained results are consistent with previous ones. The manuscript contains minor grammatical errors, lack of spaces and minor errors in references which should be corrected.

Author Response

Response to Reviewer 3 Comments

Point 1: The manuscript concerns a very significant problem in terms of mental health of adolescents. It has been properly formatted, divided into parts in accordance with the requirements of the journal. The manuscript conteins correctly selected and up-to-date literature. However, it contains a few weak points.

Response 1: Thank you for your generous work and positive suggestions, which have definitely contributed to improving our manuscript. We have studied your comments carefully and have revised the manuscript accordingly. Our point-by-point responses to the comments are listed below.

Point 2: In the introduction, the Authors should explain what opioids are abused and the reason for the increase in their use. Especially that they emphasized that adolescents use them and even abuse.

Response 2: We truly thank you for your comments and constructive suggestions. According to your suggestion, we have explained what opioids are abused and the reason for the increase in their use in the introduction (please see lines 55)

Point 3: Are these opioids so easily available in their country? This part of the manuscript is particularly controversial. How is it possible that even 12 year olds use them?

Response 3: We truly thank you for your comments and constructive suggestions. Our previous study has showed that in China, the nonmedically used prescription opioids or sedatives were most commonly obtained from home (45.0%), followed by from peers (25.9%), others (21.8%) and nightclub/pub (0.8%) among adolescents[1].

Furthermore, Table 1 shows that 1.4% and 0.7% of the 12 years old adolescents admitted that they were ‘frequent user’ of lifetime or past-year NMPOU, respectively. These results also indicated that the government, families and schools should be informed of the adverse effects of NMPOU, and government should strengthen regulations to limit the sales of prescription drugs to adolescents.

Table 1. Lifetime or past year prevalenc of nonmedically used prescription opioids

among 12 years old adolescents(N=11,552)

Variable

N(%)

Lifetime NMPOU

Abstainers

10993(95.2)

Experimenters

395(3.4)

Frequent users

164(1.4)

Past-year NMPOU

Abstainers

11254(97.4)

Experimenters

212(1.8)

Frequent users

86(0.7)

Point 4: Table 1 contains p-values but is not clearly indicated to which group the comparisions were made. This should be completed since it makes difficult to understand the results.

Response 4: We truly thank you for kind suggestions. We are sorry for our unclear description and we have added the related interpretation in the annotation of Table 1(please see lines 174)

Point 5: Table 2: the description should be supplemented with the abbreviation cOR.

Response 5: We thank you for your comments. And the abbreviation cOR was in the annotation of Table 2(please see lines 186)

Point 6: Paragraphs 3.2 and 3.3 are rather a description of the tables than the results, this should corrected.

Response 6: We truly thank you for your valuable comments. We have revised the description of Paragraphs 3.2(please see lines 178) and 3.3 (please see lines 192)

Point 7: In discussion informations in lines 236 to 241 have no direct relation to the topic. In my opinion they should be omitted. Additional manuscript weaknesses are repetitions of the same thread in discussion.

Response 7: We truly thank you for your kind suggestions. We agree with your constructive suggestions and we have deleted the lines 236 to 241(please see lines 248).

Point 8: The manuscript should be also supplemented with a description of the effect of opioids on sleep and sleep disorders.

Response 8: We truly thank you for your kind suggestions. According to your suggestion, we have added this part (please see lines 60).

Point 9:The authors should also indicate what is a novelty of this work, since they themselves clearly emphasize that obtained results are consistent with previous ones.

Response 9: We truly thank you for your kind suggestions. In our study, novelty of this work is as follows: 1) Data from individual studies on the sex differences in sleep disturbance are not entirely consistent. With the large sample, our result demonstrates a higher prevalence of sleep disturbance among girls compare to boys (please see lines 220). 2) our study firstly provides some evidence that compared with experimental opioid misuse, more frequent of prescription opioid misuse was associated with a greater likelihood of reporting sleep disturbance after adjusting for covariates(please see lines 248). 3) to our knowledge, the effect of the sex on the association of lifetime or past-year NMPOU with sleep disturbance has not specifically reported previously. This is the first study aimed at determining a sex-differential association among Chinese adolescents (please see lines 277).

Point 10: The manuscript contains minor grammatical errors, lack of spaces and minor errors in references which should be corrected.

Response 10: We truly thank you for your kind suggestions. We have revised minor grammatical errors in our previous manuscript. Furthermore, we have corrected the reference(please see lines 311). Many thanks again.

Reference

Guo, L.; Luo, M.; Wang, W.; Xiao, D.; Xi, C.; Wang, T.; Zhao, M.; Zhang, W.-H.; Lu, C., Association between nonmedical use of opioids or sedatives and suicidal behavior among Chinese adolescents: an analysis of sex differences. Australian & New Zealand Journal of Psychiatry 2019, 53, (6), 559-569.
